# Preoperative Biopsychosocial Assessment and Length of Stay in Orthopaedic Surgery Admissions of Youth with Cerebral Palsy

**DOI:** 10.3390/bs13050383

**Published:** 2023-05-06

**Authors:** Nancy Lennon, Carrie Sewell-Roberts, Tolulope Banjo, Denver B. Kraft, Jose J. Salazar-Torres, Chris Church, M. Wade Shrader

**Affiliations:** Nemours Children’s Health, Wilmington, NC 19803, USA; nancy.lennon@nemours.org (N.L.); carrie.sewellroberts@nemours.org (C.S.-R.); tolulope.banjo1@gmail.com (T.B.); denverkraft@gmail.com (D.B.K.); jose.salazar@nemours.org (J.J.S.-T.); wade.shrader@nemours.org (M.W.S.)

**Keywords:** cerebral palsy, biopsychosocial assessment, orthopaedics

## Abstract

Caregivers of children with cerebral palsy (CP) experience stress surrounding orthopaedic surgery related to their child’s pain and recovery needs. Social determinants of health can affect the severity of this stress and hinder health care delivery. A preoperative biopsychosocial assessment (BPSA) can identify risk factors and assist in alleviating psychosocial risk. This study examined the relationship between the completion of a BPSA, hospital length of stay (LOS), and 30-day readmission rates for children with CP who underwent hip reconstruction (HR) or posterior spinal fusion (PSF). Outcomes were compared with a matched group who did not have a preoperative BPSA. The BPSA involved meeting with a social worker to discuss support systems, financial needs, transportation, equipment, housing, and other services. A total of 92 children (28 HR pairs, 18 PSF pairs) were identified. Wilcoxon analysis was statistically significant (*p* = 0.000228) for shorter LOS in children who underwent PSF with preoperative BPSA (median = 7.0 days) vs. without (median = 12.5 days). Multivariate analysis showed that a BPSA, a lower Gross Motor Function Classification System level, and fewer comorbidities were associated with a shorter LOS after both PSF and HR (*p* < 0.05). Identifying and addressing the psychosocial needs of patients and caregivers prior to surgery can lead to more timely discharge postoperatively.

## 1. Introduction

### Background

Cerebral palsy (CP) is the most common pediatric motor disability, affecting an estimated 1 in 345 children in the United States [1]. The neurological pathology of CP involves a brain injury or disruption occurring before birth, during delivery, or in early childhood [2]. Motor disability in children with CP is described using a five-level classification system, the Gross Motor Function Classification System (GMFCS) [3]. Symptoms of CP differ among children, ranging from minor motor difficulties to major physical disabilities that negatively impact independent movement. Often, individuals with CP have co-occurring medical conditions and developmental disabilities alongside the musculoskeletal impairments [4].

The experience of parenting a child with CP can be stressful due to the child’s need for support in activities of daily living, need for ongoing therapies, educational advocacy, higher financial strain, and barriers to caregivers maintaining employment [5,6,7,8,9]. While every parent/caregiver of a person with CP experiences some level of stress that is higher (on average) than a parent of a typically developing child [6,10], this level of stress is greater for parents with other risks associated with social determinants of health (SDOH) [11,12]. These determinants may include financial hardship, housing and food insecurity, lack of access to quality education, environmental and neighborhood safety concerns, racism, ableism, discrimination based on sexual identity, and lack of employment opportunity [13].

Children and adolescents with CP often require orthopaedic surgery to maintain or improve their function, reduce pain, and improve quality of life [14,15,16,17,18]. Common orthopaedic surgeries for children functioning at GMFCS levels IV and V include posterior spinal fusion (PSF) and hip reconstruction (HR) [17,18]. The experience of surgery, a hospital stay, rehabilitative therapies, and recovery is stressful for both patients and their parents/caregivers [19,20]. The stress caused by these surgical events is multi-faceted: concern about the risks and long-term outcomes of surgery, the experience of pain and recovery from the surgical procedure itself, the financial strain of a hospitalization, disruption to daily routine and parent employment, and childcare concerns for siblings [21].

If medical teams can evaluate the family system’s SDOH and baseline stress load, the team may be able to support the family in minimizing these factors and making the stress of a surgical event more manageable. One avenue for capturing the SDOH of families is through the administration of a biopsychosocial assessment (BPSA), a tool utilized by medical social workers to develop treatment and intervention plans [22]. A BPSA reveals a family’s basic composition; strengths such as resilience, family/community support, and health literacy; as well as areas of risk, including SDOH barriers and access to needed therapies, benefits, and services.

The aim of this study was to examine the relationship between the completion of a preoperative BPSA and hospital discharge metrics including length of stay (LOS) and 30-day readmission rate (RR) for children with CP undergoing PSF or HR surgeries.

## 2. Materials and Methods

This Institutional-Review-Board-approved retrospective cohort study included children with CP who underwent hip or spine surgery. Potential cases were identified from a historical database from the authors’ institution. Inclusion criteria were (a) diagnosis of CP classified at GMFCS levels IV and V, (b) underwent PSF or HR at the authors’ institution between 2017 and 2021, and (c) aged 2 to 21 years. Children whose families completed a BPSA were selected from this group and were then matched according to surgery type (hip or spine), age (within 2 years), number of comorbid conditions, and GMFCS level to a group that did not have a preoperative BPSA. Comorbidities were categorized as seizures, gastrostomy tube, tracheostomy, or non-verbal [23]. Based on chart review, the number of comorbidities for each patient was identified and, for statistical analysis, was ranked as none = 0, small = 1 or 2, and large = 3 or 4.

### 2.1. Biopsychosocial Assessment

The BPSA utilized by the social work team was developed based on the guidelines set by the National Association of Social Workers Standards for Social Work Practice in Health Care Settings [24]. It includes the assessment of patient and caregiver strengths, such as self-efficacy, access to family, faith and community supports and resources, and resilience. The BPSA also identifies SDOH risks such as transportation, food, housing, employment, income, access to government benefits, homecare services, home accessibility, access to needed durable medical equipment, access to therapies, and mental health care (see the Appendix A). The BPSA was utilized as a guide for the medical social worker to provide interventions for the family to mitigate the stress burden of specific areas of risk. For example, if the BPSA identified that a family had housing insecurity, the social worker would assist the family in contacting the state housing authority to check housing vouchers. If the BPSA identified that the family had transportation barriers ahead of the surgery, the social worker would assist the family in scheduling Medicaid transportation for preoperative appointments and on the day of the surgery. If the family identified a need for mental health therapy to cope with stress, anxiety, or depressive symptoms, the social worker would assist the family in accessing mental health care. The roll out of the BPSA in late 2018 was controlled by the CP division chief, beginning with his own patients, and limited in scope by social work resources. The referral process for BPSA was formalized and social work resources were increased in 2019. By 2020, all children (GMFCS IV and V preoperative for HR PSF) were referred for preoperative BPSA.

Primary outcome variables included postoperative LOS (number of days), rate of extended LOS (ELOS), and 30-day RR. Length of stays over the median (6 days for HR and 10 days for PSF) were considered ELOS. For any child with a readmission within 30 days, a chart review was performed to determine reasons for readmission.

### 2.2. Statistical Analysis

Chi-squared analysis was completed to examine differences in matching criteria between BPSA and no-BPSA groups. The median LOS, rate of ELOS, and 30-day RR for each type of surgery (hip or spine) were compared between BPSA and no-BPSA groups. Statistical analyses were carried out with a Wilcoxon test for LOS and a chi-squared analysis for ELOS and 30-day RR.

A general linear regression model (GLM) with a Poisson distribution and a stepwise function to select relevant variables was used to predict LOS in days for each type of surgery. Variables in the model included BPSA (yes, no), number of comorbidities (none = 0, small = 1 or 2, large = 3 or 4), age, sex, and GMFCS level. Similarly, a GLM with a binomial distribution and a stepwise function to select relevant variables was used to determine the effect of these same factors on whether LOS was within the median range (≤6 days for HR, 10 days for PSF) or extended (>6 days for HR, >10 days for PSF). An additional GLM with a binomial distribution and a stepwise function to select relevant variables was used to determine whether these factors influenced 30-day RR. All statistical analyses were performed using R [25]. Significance level for all tests was set at *p* < 0.05.

## 3. Results

Forty-six children with CP who had a BPSA were matched with forty-six children who did not have a BPSA with similar age, GMFCS level, and number of comorbidities (Table 1). Fifty-six children had HR and thirty-six had PSF. Table 2 shows the LOS median, interquartile range, confidence interval, and 30-day readmission (*n* = 10) for the 92 children included in this analysis and the distribution according to preoperative BPSA, type of surgery, and number of comorbidities.

### 3.1. Group Analysis

#### 3.1.1. Length of Stay

The median LOS for the children in the PSF group was 10 days. The difference in LOS in this group was statistically significant between the BPSA (median [CI] = 7.0 [1.4]) and no-BPSA groups (median [CI] = 12.5 [12.2]; *p* = 0.00023). The median LOS for the children in the HR group was six days. After HR, there was no significant difference in LOS between the BPSA (median [CI] = 6.0 [3.7]) and no-BPSA groups (median [CI] = 7.0 [1.5]) (*p* = 0.51; Figure 1).

#### 3.1.2. Extended Length of Stay

Three of the eighteen children who had a BPSA in the PSF group had an ELOS, compared with twelve out of eighteen children who did not have a BPSA. Chi-squared analysis showed a significant difference between these groups (*p* = 0.0023). In the HR group, 3 out of 28 (BPSA) and 7 out of 28 patients (no BPSA) had an ELOS (*p* = 0.16).

#### 3.1.3. Thirty-Day Readmission

Three of the eighteen children in the PSF surgery group who had a BPSA were readmitted within 30 days, while one of the eighteen children who did not have a BPSA was readmitted within 30 days (*p* = 0.29). One of the twenty-eight children in the HR surgery group who had a BPSA was readmitted within 30 days and five of the twenty-eight children who did not have a BPSA were readmitted within 30 days (*p* = 0.084). Reasons for 30-day readmission can be found in Table 3.

### 3.2. Multivariate Analyses

#### 3.2.1. Length of Stay

In the LOS model for PSF, statistically significant effects were found for the following factors: the inclusion of a preoperative BPSA was associated with a shorter LOS (*p* < 0.001); an additional number of comorbidities, both small (1,2) (*p* = 0.03) and large (3,4) (*p* < 0.001), was associated with a longer LOS; and a higher GMFCS level was associated with a longer LOS (*p* = 0.005). Older children tended to have an increased LOS, but this did not reach significance levels (*p* = 0.06).

In the LOS model for HR, statistically significant effects were found for the following factors: an additional number of small medical issues was associated with a shorter LOS (*p* < 0.001), and a longer LOS was observed for male children (*p* = 0.019). For this group, the inclusion of a preoperative BPSA was not relevant to the LOS (*p* > 0.05). Table 4 shows the summary of the multivariate analysis for children who underwent PSF and HR.

#### 3.2.2. Extended Length of Stay

For children who underwent PSF, the ELOS model was statistically significant for a median or shorter LOS in patients with a BPSA (*p* = 0.004). For children who underwent HR, the ELOS model found statistically significant effects for BPSA and a median or lower LOS (*p* = 0.03), higher GMFCS level, and extended LOS (*p* = 0.04). Male patients tended to have an ELOS but this did not reach significance levels (*p* = 0.05). Table 5 shows the summary of the multivariate analysis for children who underwent PSF and hip reconstruction.

#### 3.2.3. Thirty-Day Readmission

There were no statistically significant relationships between any of the factors included in this study and a 30-day RR model for either the PSF or HR groups.

## 4. Discussion

Family stress associated with caring for a child with CP can be exacerbated by orthopaedic surgery due to pain and financial impact [19,20,21]. Risks associated with social determinants of health can increase caregiver stress and lead to difficulties in caring for a child with CP [11,12]. Identification of SDOH utilizing a BPSA facilitates the implementation of psychosocial interventions aimed at reducing risk and improving outcomes in care [22].

In late 2018, our hospital undertook a new initiative aimed at reducing disparities in health outcomes for youth with CP undergoing orthopaedic surgery. The social work and orthopaedic teams led a program to offer psychosocial support to families in this group. During the first year of the program, referrals for BPSA were low relative to the number of eligible families and some families did not receive the assessment. By 2021, nearly all patients having orthopaedic surgery with an anticipated hospital LOS of more than a few days completed a BPSA prior to surgery. The time frame in which the service was developing allowed us to examine the impact of the BPSA on hospital admissions.

Average hospital LOS for patients with PSF was significantly shorter for those who completed a BPSA compared with those who did not. For patients with HR, there was no difference in average LOS for those who completed a BPSA compared with those who did not. Examining LOS on a dichotomous scale, as extended (>median) or not (≤median), revealed similar results with a significant difference between BPSA and no-BPSA groups in patients who had PSF but not HR. Children tend to have longer admissions after PSF than with HR due to the greater burden of the surgery. Perhaps with this greater burden, more demand was placed on family resources, leading to a stronger (statistically significant) impact of the BPSA. Anecdotally, nursing staff and case managers report that families with a BPSA experience smoother, less chaotic hospital discharge.

The 30-day RR for the 92 patients in this analysis was 9.8% and was not statistically different between those who had a BPSA and those who did not. We observed a trend of fewer 30-day readmissions in HR patients who had a BPSA, though there was no significant difference (4% vs. 18% in patients without a BPSA). We continue to observe this trend in our clinical practice and expect to find significant differences as we analyze larger groups and examine reasons for 30-day readmissions with standardized methods.

The medical complexity of youth with CP functioning at GMFCS levels IV/V justified additional multifactorial analysis of LOS after PSF and HR. Studying a similar patient sample, Jain et al. reported a higher frequency of complications following PSF in those with more comorbidities: 49% in those with three or four comorbidities compared with 12% in those with no comorbidities [23]. When we included factors that capture medical comorbidities and motor disability, we found statistically significant results. Multifactorial analysis revealed that a preoperative BPSA, a lower GMFCS level (IV vs. V), and fewer comorbidities were associated with a shorter median LOS and less frequent ELOS following PSF and HR. A higher GMFCS level and higher number of comorbidities likely contribute to medical complications that extend hospital stay, but these same factors can also result in higher caregiving demands and stress, leading to a possible psychosocial explanation for delays in discharge.

Medical chart review for this study tried to identify explanations for ELOS and readmission within 30 days. Reasons for 30-day readmission and ELOS offered a mix of medically driven diagnoses such as infection, patient management difficulties such as postoperative pain and constipation, as well as combinations of medical/social circumstances. While it was clear that some issues could have been addressed with preventative strategies identified through a preoperative BPSA, we could not analyze reasons for ELOS and 30-day RR statistically in this work.

This study serves to capture our new clinical practice of completing a BPSA preoperatively in children with CP undergoing orthopaedic surgery who have an anticipated hospital stay of more than a few days. Given the benefit demonstrated, we advocate for clinicians in similar settings to institute this process. The BPSA standardizes the process of identifying SDOH. Our current clinical practice aims to develop methods qualifying improvement in issues identified through the preoperative BPSA. While the social work teams utilized the BPSA to identify areas of risk and guide the implementation of patient- and family-specific interventions, some issues identified through the BPSA (for example, chronic financial insecurity) may be beyond the scope of the relatively short time frames of preoperative planning and preparation. We are working with the Health Equity Office at our hospital to identify strategies to address these broader issues.

Limitations of this study include the analysis of only PSF and HR surgical groups. While these two procedures are common and highly impactful, both in health benefit to the child and reduction in caregiving needs, there are other common surgery situations in which families likely benefit but whom we did not study. The retrospective nature of the study introduces the possibility of bias in the study groups. While we tried to minimize this in our methodology, there could be differences in the early referral patterns that contributed to group differences. Some physicians were early adopters of the service, while others took longer to integrate the preoperative BPSA into their practice. We used BPSA as a marker for intervention; we did not look at the interventions and assumed that after BPSA, appropriately directed interventions were delivered. Finally, the medically complex nature of this patient population may contribute to outliers in LOS that could impact results unrelated to the BPSA.

## 5. Conclusions

Identifying and addressing psychosocial needs of patients and their caregivers through a preoperative BPSA is associated with positive impacts on hospital quality metrics including less time spent in the hospital after surgical admissions for spinal fusion for youth with CP.

## Figures and Tables

**Figure 1 behavsci-13-00383-f001:**
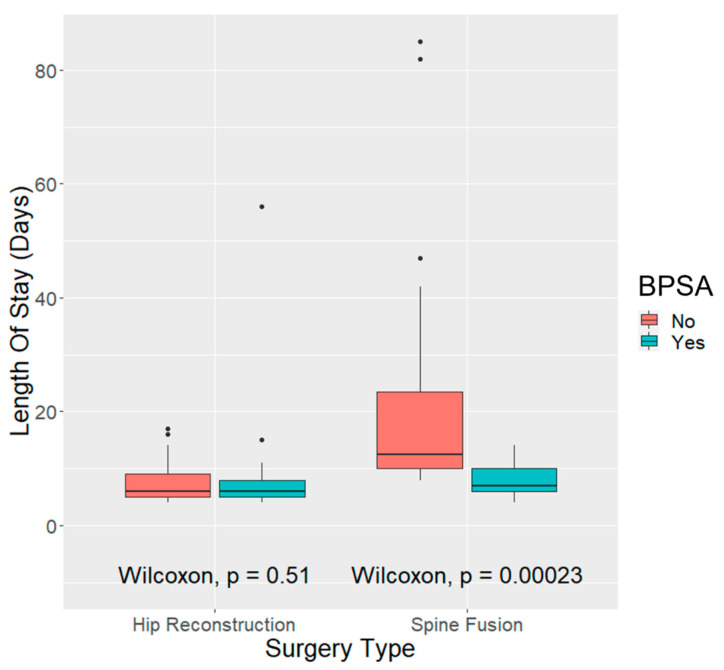
Differences in length of stay in days between biopsychosocial assessment (BPSA) and no-BPSA groups following hip reconstruction and spine fusion.

**Table 1 behavsci-13-00383-t001:** Distribution of matching variables for children who underwent spinal surgery and had preoperative biopsychosocial assessment (BPSA) or no BPSA.

Variable		BPSA		*p* Value	Test
		No	Yes		
Age	Median (CI)	10.6 (0.94)	10.9 (0.97)	0.72	Wilcoxon
Race	Asian	1	0	0.68	Chi-squared
	Asian Indian	1	0		
	Black or African American	13	13		
	Guamanian or Chamorro	0	1		
	Some other race	3	4		
	White or Caucasian	28	28		
GMFCS	IV	16	17	0.83	Chi-squared
	V	30	29		
Surgery type	Hip	28	28	1	Chi-squared
	Spine	18	18		
Range of medical issues	None = 0	6	8	0.84	Chi-squared
	Small = 1.2	24	23		
	Large = 3.4	16	15		

Note: Chi-square analyses were used to confirm that matching variables were equivalent between BPSA and no-BPSA groups. Children were not matched for race, but the distribution was similar between groups. GMFCS, Gross Motor Function Classification System.

**Table 2 behavsci-13-00383-t002:** Descriptive statistics for children who underwent hip reconstruction or posterior spinal fusion, whether they had a biopsychosocial assessment (BPSA) administered, the number of comorbidities, and whether they were readmitted within 30 days.

Surgery Type	BPSA	Comorbidities	*n*	LOS Days			30-Day Readmission
				Median	IQR	CI	
Hip	No	None	4	5.5	1.75	2.72	0
Hip	No	Small	16	6.5	4	1.76	2
Hip	No	Large	8	6.5	7.75	4.07	2
Spine	No	None	2	11	3	38.1	1
Spine	No	Small	8	11	2.5	21.2	0
Spine	No	Large	8	21	31	21.8	0
Hip	Yes	None	6	6	1.5	21.6	0
Hip	Yes	Small	16	5.5	2.25	1.06	1
Hip	Yes	Large	6	8	1.5	3.8	0
Spine	Yes	None	2	9	1	12.7	0
Spine	Yes	Small	7	10	5	3.16	1
Spine	Yes	Large	9	7	1	1.96	2

IQR, interquartile range; LOS, length of stay.

**Table 3 behavsci-13-00383-t003:** Reasons for 30-day readmission.

Subject Number	Surgery	Reason for 30-Day Readmission
1	HR	Oropharyngeal dysphagia, diabetes insipidus
2	HR	Postoperative pain management
3	HR	Vomiting, constipation
4	HR	Decubitus ulcer
5	PSF	Urinary retention
6	HR	Wound infection
7	PSF	Urinary tract infection
8	PSF	Wound infection
9	PSF	Wound infection

HR, hip reconstruction; PSF, posterior spinal fusion.

**Table 4 behavsci-13-00383-t004:** Summary of the length of stay multivariate model for children in the PSF and hip reconstruction groups.

Length of Stay Models Fit				
**PSF**				
χ^2^ (5)	186.96			
*p*	0.00			
Pseudo-R^2^ (McFadden)	0.29			
Standard errors: MLE				
	**Est.**	**S.E.**	**z val.**	** *p* **
(Intercept)	0.17	0.64	0.27	0.78
BPSA (yes)	−1.08	0.10	11.16	<0.001
additional comorbidities (small)	0.37	0.17	2.16	0.03
additional comorbidities (large)	0.63	0.19	3.27	<0.001
Age	0.05	0.02	1.90	0.06
GMFCS	0.41	0.12	3.48	0.005
**Hip Reconstruction**				
χ^2^ (3)	22.56			
*p*	0.00			
Pseudo-R^2^ (McFadden)	0.06			
Standard errors: MLE				
	**Est.**	**S.E.**	**z val.**	** *p* **
(Intercept)	2.23	0.12	19.24	0.00
Additional comorbidities (small)	−0.46	0.12	−3.89	<0.001
Additional comorbidities (large)	−0.13	0.13	−1.03	0.30
Sex (male)	0.23	0.1	2.34	0.019

BPSA, biopsychosocial assessment; GMFCS, Gross Motor Function Classification System.

**Table 5 behavsci-13-00383-t005:** Summary of the length of stay multivariate model for children in the PSF and hip reconstruction groups.

Extended Length of Stay Models Fit				
**PSF**				
χ^2^ (1)	9.77			
*p*	0.00			
Pseudo-R^2^ (McFadden)	0.20			
Standard errors: MLE				
	**Est.**	**S.E.**	**z val.**	** *p* **
(Intercept)	0.69	0.5	1.39	0.17
BPSA (Yes)	−2.3	0.81	−2.86	0.004
**Hip Reconstruction**				
χ^2^ (3)	11.66			
*p*	0.01			
Pseudo-R^2^ (McFadden)	0.22			
Standard errors: MLE				
	**Est.**	**S.E.**	**z val.**	** *p* **
(Intercept)	−13.99	6.07	−2.31	0.02
BPSA (yes)	−2.16	1.01	−2.13	0.03
Sex (male)	1.96	1.02	1.92	0.05
GMFCS	2.56	1.22	2.09	0.04

BPSA, biopsychosocial assessment; GMFCS, Gross Motor Function Classification System; MLE, maximum likelihood estimation.

## Data Availability

The data presented in this study are available on request from the corresponding author.

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
