# Peer review of "Preoperative Biopsychosocial Assessment and Length of Stay in Orthopaedic Surgery Admissions of Youth with Cerebral Palsy"

_behavsci, 2023, doi:10.3390/bs13050383_

Round 1
Reviewer 1 Report
The authors present a study of children with GMFCS 4+5 CP, undergoing spine or hip surgery, comparing those who had a social assessment to those without. Interestingly, there is no discussion of the score or results of the assessment. Rather, simply whether the assessment was performed is the tested intervention. They find significantly shorter LOS in children with the assessment.
Points for clarification:
Is the BPSA a validated measure? Why was it used rather than something like the Psychosocial Assessment Tool?
How was it decided who would receive BPSA? It appears from the Discussion that BPSA was rolled out, and therefore, this study is something like historical control. These details should be more clearly noted in the Methods, not only in Discussion.
Why consider LOS both as a continuous variable and a dichotomous variable. Which is most representative of what the researchers wish to evaluate? It might be more appropriate to focus on one or the other way to measure that concept.
Reviewer 2 Report
This is a retrospective study, with index cases being compared to historical controls. the study covers the period from 2017 to 2021 and the use of BPSA was first introduced in late 2018 and not widely used until 2020-2021 (pages 6,7) so the comparison is between children having surgery ini 2021 to those in 2017-18 so the shorter LOS in the spine surgery group (no diifference in hip surgery group) could be due to improved surgical technique or post op care over the years rather than the BPSA. It is intuitive to think the BPSA would help but this was not the case in the hip surgery cases.
The conclusion: "preoperative BPSA is associated with positive impacts on hospital quality metrics including less time spent in the hospital after surgical admissions for youth with CP." Was only true for LOS for spinal surgery and not hip surgery, so this is an overstatement even of the data as presented, independent of the retrospective aspect of the study.
Also the authors can not refer to differences that are not significantly different: if not significantly different then they are not different! e.g. "The difference in LOS between BPSA (median [CI] = 6.0 [3.7]) 138 and no BPSA groups (median [CI] = 7.0 [1.5]) was not significant (p=0.51; Figure 1). "
This should be: "there was no significant difference in LOS between...."
Or: "The difference between these groups approached significance (p=0.084)."
Close does not count. This is an overstatement of the data
Round 2
Reviewer 2 Report
The authors addressed most, but not all of the concerns about non-signiificant results:
"Male patients tended to have an ELOS but this did not reach significance levels (p=0.05)"
section 3.2.2. Can't talk about tending. Need to say: Male patents did not have a significantly different ELOS (p=0.05"
In discussion:
"We observed a trend of 218 fewer 30-day readmissions in HR patients who had BPSA though there was no significant 219 difference(4% vs 18% in patients without BPSA). We continue to observe this trend in our 220 clinical practice and expect to find significant differences as we analyze larger groups and 221 examine reasons for 30-day readmissions with standardized methods."
Can't do this. What they CAN do is have, in the discussion, a comment on study limitations, which would be that the study was not sufficiently powered to detect these differences.